# Prognostic Role of *KRAS G12C* Mutation in Non-Small Cell Lung Cancer: A Systematic Review and Meta-Analysis

**DOI:** 10.3390/diagnostics13193043

**Published:** 2023-09-25

**Authors:** Durgesh Wankhede, Christophe Bontoux, Sandeep Grover, Paul Hofman

**Affiliations:** 1Division of Clinical Epidemiology and Aging Research, German Cancer Research Center (DKFZ), 69120 Heidelberg, Germany; 2Laboratory of Clinical and Experimental Pathology, Pasteur Hospital, Centre Hospitalier, Université Côte d’Azur, 06002 Nice, France; bontoux.c@chu-nice.fr; 3Centre for Genetic Epidemiology, Institute for Clinical Epidemiology and Applied Biometry, University of Tübingen, 72076 Tübingen, Germany; grover@staff.uni-marburg.de; 4Institute for Research on Cancer and Ageing, Nice (IRCAN), INSERM U1081 and UMR CNRS 7284, Team 4, 06107 Nice, France; hofman.p@chu-nice.fr; 5Hospital-Integrated Biobank BB-0033-00025, Pasteur Hospital, 06000 Nice, France; 6University Hospital Federation OncoAge, CHU de Nice, University Côte d’Azur, 06000 Nice, France

**Keywords:** non-small cell lung cancer, NSCLC, *KRAS*, *KRAS G12C*, systematic review, meta-analysis

## Abstract

*KRAS G12C* mutation (*mKRAS G12C*) is the most frequent *KRAS* point mutation in non-small cell lung cancer (NSCLC) and has been proven to be a predictive biomarker for direct *KRAS G12C* inhibitors in advanced solid cancers. We sought to determine the prognostic significance of *mKRAS G12C* in patients with NSCLC using the meta-analytic approach. A protocol is registered at the International Prospective Register for systematic reviews (CRD42022345868). PubMed, EMBASE, The Cochrane Library, and Clinicaltrials.gov.in were searched for prospective or retrospective studies reporting survival data for tumors with *mKRAS G12C* compared with either other *KRAS* mutations or wild-type *KRAS* (*KRAS*-*WT*). The hazard ratios (HRs) for overall survival (OS) or Disease-free survival (DFS) of tumors were pooled according to fixed or random-effects models. Sixteen studies enrolling 10,153 participants were included in the final analysis. *mKRAS G12C* tumors had poor OS [**HR, 1.42; 95% CI, 1.10–1.84, *p* = 0.007**] but similar DFS [HR 2.36, 95% CI 0.64–8.16] compared to *KRAS-WT* tumors. Compared to other *KRAS* mutations, *mKRAS G12C* tumors had poor DFS [**HR, 1.49; 95% CI, 1.07–2.09, *p* < 0.0001**] but similar OS [HR, 1.03; 95% CI, 0.84–1.26]. Compared to other *KRAS* mutations, high PD-L1 expression (>50%) [**OR 1.37 95% CI 1.11–1.70, *p* = 0.004**] was associated with *mKRAS G12C* tumors. *mKRAS G12C* is a promising prognostic factor for patients with NSCLC, negatively impacting survival. Prevailing significant heterogeneity and selection bias might reduce the validity of these findings. Concomitant high PD-L1 expression in these tumors opens doors for exciting therapeutic potential.

## 1. Introduction

Approximately 236,000 new lung cancer cases are expected to be diagnosed in the United States in 2022, contributing to 12.3% of all new cancer cases and 21.4% of all cancer-related deaths [1]. Of these, 85% are non-small cell lung Cancer (NSCLC), and 75% of NSCLC cases present at either the advanced or relapsed stage [2]. Kirsten rat sarcoma viral oncogene homolog (*KRAS*) is one of the most common oncogenic drivers in NSCLC, seen in over 30% of lung adenocarcinomas (LUAD), depending on ethnicity and tumor stage and associated with smoking and female patients [3,4,5]. Notwithstanding this, *KRAS* testing is not included in the routine genomic panel for NSCLC, probably due to its less well-established efficacy in daily clinical practice based on current evidence [The European Society for Medical Oncology (ESMO) Scale for Clinical Actionability of Molecular Targets (ESCAT) recommendation category 2B] [6,7]. Despite exhaustive exploration, the prognostic role of *KRAS* status remains contentious [8,9,10,11].

*KRAS* status harbors a spectrum of distinctly mutated *KRAS* substitutions [12]. In NSCLC, codon 12 is the most common site for *KRAS* point mutations, and glycine-to-cysteine (*mKRAS G12C*) (40–50% of all *KRAS* mutations) is present in 10–13% of LUAD patients [13,14]. Other common point mutations are glycine-to-aspartic acid (*mKRAS G12D*) and glycine-to-valine (*mKRAS G12V*) substitutions, which are observed in approximately 5% and 4% of LUAD patients, respectively [12]. Epidemiologically, *mKRAS G12C* and *mKRAS G12V* are associated with a history of smoking, whereas *mKRAS G12D* is associated with non-smokers [15,16]. These mutations can uniquely alter downstream effector molecules, leading to mutation-specific signal transduction, which can eventually modify clinical outcomes and treatment responses [17,18]. Molecular dynamics studies have shown that *KRAS* mutant proteins differ from wild-type proteins and have mutation subtype-specific differences [19].

Recently, leveraging the peculiar structural and biochemical properties of *mKRAS G12C*, various selective *KRAS G12C* small-molecule inhibitors have been developed [20]. The clinical efficacy of these newer agents is being explored in multiple clinical trials for advanced solid tumors [21,22,23]. Direct *KRAS G12C* inhibitors, sotorasib and adagrasib, received accelerated approval from the U.S. Food and Drug Administration (FDA) for previously treated advanced NSCLC patients with *mKRAS G12C* [24,25]. Similarly, based on the results of the CodeBreak200 trial (*n* = 345), ESMO guidelines recommend sotorasib as second-line therapy for advanced NSCLC patients with *mKRAS G12C* [26,27]. Based on its relatively ubiquitous epidemiology and distinctive biological behavior, this study aimed to quantitatively synthesize evidence on the prognostic role of *mKRAS G12C* in patients with NSCLC. Specifically, the primary aim of this study was to evaluate whether the survival rate of NSCLC patients with *mKRAS G12C* is similar to that of patients with tumors with either other *KRAS* mutations or *KRAS-wild type* (*KRAS-WT)*. The secondary aim was to determine the association between frequent *KRAS*-comutations and *mKRASG12C* tumors. In addition, the correlation between PD-L1 expression levels and the *KRAS* subtype was investigated.

## 2. Materials and Methods

### 2.1. Search Strategy and Study Selection

Preferred Reporting Items for Systematic Review and Meta-analysis (PRISMA) guidelines were followed to conduct this study [28] (Appendix A). The protocol was registered in the International Prospective Register of Systematic Reviews (PROSPERO identifier: CRD42022345868). A systematic and comprehensive search was conducted by two reviewers (DW and PH) of PubMed, Embase, Cochrane Library, and ClinicalTrials.gov from inception until April 2023. The following combinations of keywords and Medical Subject Headings (MeSH)/EMTREE terms were used: ‘*KRAS* OR Kirsten rat sarcoma viral homolog’, ‘mutation* OR mutated’, and ‘lung* OR pulmonary’, ‘cancer* OR tumor* or tumour* OR carcinom* OR neoplas* or malignan*’ and ‘prognos* or survival or recurren* or mortality or predict* or outcome* or death’. The search was restricted to articles published in English.

Inclusion criteria for eligible studies were defined as follows:(a)Studies investigating adult patients (>18 years) with pathologically confirmed NSCLC;(b)Studies assessing the mutation status of the *KRAS* gene in primary lung cancer tissue;(c)Studies reporting time-to-event data, including overall survival (OS), and disease-free survival (DFS) for the individual groups of interest, i.e., *KRAS G12C*, other point mutations of *KRAS* gene and *KRAS-WT*;(d)Original studies, including randomized-controlled (RCT) and non-randomized-controlled (NRCTs) studies, enrolling more than five individuals in the *KRAS G12C* group;(e)Studies published in English in a peer-reviewed journal.

The exclusion criteria for the study selection were as follows:(a)Studies investigating small cell lung cancers;(b)*KRAS* mutation analysis performed in plasma;(c)Studies reporting mixed data for *KRAS G12C* and other *G12* substitutions;(d)Conference abstracts and preprint articles;(e)Narrative and systematic reviews, meta-analyses, expert opinions, editorial letters, case reports, and case series with a sample size of fewer than five individuals.

Studies exported from the databases were deduplicated, and unique studies were screened based on title, keyword, and abstract information. Subsequently, eligible studies were screened for full-text information, and data extracted from the studies met the eligibility criteria.

### 2.2. Data Extraction and Quality Assessment

Two reviewers independently retrieved the following data: authors, patient source, study design, histology, stage, mutation analysis methods, the total number of patients and patients with *KRAS G12C* mutation, and median follow-up duration in patients with *KRAS WT*, *mKRAS G12C*, other *KRAS* mutations. In addition, statistics extraction for time-to-event analyses (OS and DFS) were Hazard Ratio (HR) and 95% Confidence Interval (CI), median survival, and *p*-values. The OS and DFS were defined per the definition in each primary study. In case of a lack of reported survival outcomes, data were reconstructed from Kaplan–Meier curves using Tierney’s or Parmar’s methods [29,30]. We also assessed whether the survival outcomes were adjusted for clinicopathological covariates through univariate or multivariate analysis. We would utilize the latter if the author reported univariate and multivariate survival analysis results. Items were listed as “not available” (NA) when data from any of the categories mentioned above was unavailable”.

The publication that provided the most recent or informative data for studies with multiple publications was selected. Discrepancies between reviewers were resolved by consensus or involving a third investigator. Survival outcomes were compared between *KRAS G12C* vs. other *KRAS* mutations or *KRAS G12C* vs. *KRAS-WT*. In studies where survival data for individual point mutations were provided, we compared *KRAS G12C* with *KRAS G12D* mutation. The risk of bias assessment was done using the Newcastle–Ottawa Scale (NOS) for observational studies [31].

### 2.3. Statistical Methods

The pooled HR evaluated the prognostic role of *KRAS G12C* for OS and DFS using the generic inverse variance method. The statistical heterogeneity within studies was tested with the Cochrane Q test and measured using I^2^ indices. If HRs were found to have mild (I^2^ ≤ 30%) to moderate (I^2^ = 30–60%) heterogeneity, a fixed-effects model was used. In case of significant heterogeneity (I^2^ > 60%), a random-effects model was used. By convention, an observed HR > 1 implies worse survival for the group with *KRAS G12C* mutations. The impact of *KRAS* status on survival was considered statistically significant if the 95% CI did not overlap with 1. Statistical significance was set at *p* < 0.05, and all tests were two-sided. Sensitivity analyses were conducted using the leave-one-out method to determine the undue influence of an individual study on the summary estimate or heterogeneity.

We conducted subgroup analyses for outcomes with ≥5 studies and >1 study in each subgroup [32]. The studies were stratified according to ethnicity (Asian/non-Asian), testing methodology (next-generation sequencing (NGS)/polymerase-chain-reaction (PCR)), and adjustment for clinical covariates (HR derived from multivariate (MV)/univariate or recreated from survival curves (UV)). In order to decrease the likelihood of chance differences arising from multiple testing in the subgroup analyses, we used 99% CI for the study estimates and 95% CI for the summary estimate.

Pooled Odds Ratio (OR) with 95% CI for binary variables and standardized mean difference for continuous variables were generated to investigate the relationship of mutant *KRAS G12C* tumors with PD-L1 expression status (<1%, 1–50%, >50%) and co-occurring mutations (*TP53*, *STK11*).

Publication bias was assessed by Begg’s funnel plots and Egger’s test [33]. In case of publication bias, the trim-and-fill method was used to determine an adjusted pooled estimate [34]. Primary analyses were performed using the Review Manager version 5.4 (The Cochrane Collaboration, Copenhagen, Denmark), and publication bias evaluation was performed in the JASP software (JASP 0.16, the JASP team) [35].

## 3. Results

### 3.1. Study Selection and Characteristics

Eight hundred ninety-nine relevant studies were identified from electronic databases, and forty studies were selected for the full-text assessment. A manual search of the references did not yield any relevant study. Finally, sixteen studies were eligible for the systematic review, which were published between 2013 and 2021 [36,37,38,39,40,41,42,43,44,45,46,47,48,49,50,51]. The study selection process is presented in Figure 1.

The study characteristics are reported in Table 1. The total number of patients was 10,153 (84 to 2055), with the median age ranging from 61 to 69. Female patients represented 36.5% of the pooled study population. Three studies were conducted in Asia, representing 8.7% of the pooled study population [38,44,48]. Eight studies reported data specifically on advanced stages (IIIb-IV), representing approximately 49.7% of the study population [36,38,39,41,44,45,50,51]. Seven studies exclusively included non-squamous NSCLC [40,41,42,43,47,49,50]. Next Generation Sequencing (NGS) was the predominant method used for KRAS mutation analysis. The median follow-up duration ranged from 9 to 95 months. In six studies, survival data were retrieved from survival curves to rebuild the HR estimates and their variance [36,38,39,40,42,45]. Ten studies reported HRs derived from multivariate analyses, and most of these studies adjusted for covariates such as age, sex, smoking status, stage, and various treatment modalities.

In the overall NSCLC cohort, the rate of KRAS G12C mutation was 11.7%, whereas it was 40.9% in the mutant KRAS (mKRAS) population. Among the studies that reported KRAS-comutations, the frequency of TP53, STK11, and KEAP1 mutation ranged from 25–85%, 8–30%, and 5–23%, respectively. The information on PD-L1 expression status was available for 28–100% of tumor samples from five studies. In the mKRAS cohort, the frequency of tumors with negative (<1%), positive (>1%), and high PD-L1 expression (>50%) was 35% (7.6–49%), 41.2% (39–76%), and 33% (27–38.3%), respectively. Similarly, in the mKRAS G12C cohort, PD-L1 expression was found to be negative in 34% (3–47%) of the study participants, positive in 63% (45–90.5%), and high in 40.6% (18.6–47.6%).

### 3.2. Quality of Studies

The Newcastle-Ottawa Scale indicated that nine studies had a low risk of bias, and the remainder had a moderate risk of bias (Table 1 and Appendix A). The overall quality of studies was moderate to high, with an average score of 7 (range 5–9). A few studies failed to report the duration and adequacy of follow-up. In a few studies, comparability between the two groups could not be ascertained due to the lack of adjustment of confounding variables that were likely to affect the survival outcomes.

### 3.3. Prognostic Role of KRAS G12C

#### 3.3.1. KRAS G12C versus Other KRAS Mutations

Fourteen studies with 4352 patients were included in the primary meta-analysis for OS [36,37,38,39,40,41,42,43,44,45,46,49,50,51]. Five studies with 3140 patients were included in the meta-analysis for DFS.

The summary HR for OS showed no statistically significant survival difference between patients with KRAS G12C and non-KRAS G12C mutations [HR 1.03, 95% CI: 0.84–1.26, *p* = 0.79], though between-study heterogeneity was high (I^2^ = 68%, *p* < 0.0001) (Figure 2A)

Studies that compared *KRAS G12C* to *KRAS G12D* mutated tumors also showed similar outcomes [HR, 0.93, 95% CI, 0.67–12.9; *p* = 0.66] (Appendix A). Sensitivity analysis did not identify any undue influence of individual studies on effect size or heterogeneity. An asymmetrical right-skewed funnel plot (Appendix A) and significant Egger’s test (*p* = 0.013) suggested the presence of publication bias. The trim-and-fill method led to the addition of three studies; however, the combined HR after adjustment remained statistically non-significant [adjusted HR (aHR) = 0.92 95% CI 0.73–1.17] (Appendix A).

Studies were stratified according to the patients’ sources, testing methods, and survival data sources. The combined HR for Asians was 0.64 (95% CI 0.07–5.83, *p* = 0.69), and for non-Asians was 1.07 (95% CI 0.89–1.28, *p* = 0.47) (Appendix A). Among the studies with NGS-based testing methods, the combined HR was 0.73 (95% CI 0.49–1.07, *p* = 0.11), whereas, for PCR-based methods, it was 1.20 (95% 0.71–2.01, *p* = 0.50) (Appendix A). Finally, studies with HRs derived from multivariate analyses [pooled HR = 0.91 95% CI 0.59–1.42, *p* = 0.68] and HRs derived from univariate analyses/extracted from survival curves [pooled HR = 1.05 95% CI 0.91–1.22, *p* = 0.29] had non-significant results (Appendix A).

We conducted analyses on studies that included non-squamous NSCLC exclusively and found similar OS in patients with *mKRAS G12C* and other *KRAS* mutations [HR = 1.06 95% CI 0.81–1.40, *p* = 0.66] (Appendix A). Furthermore, an analysis of studies with advanced-stage NSCLC found a comparable outcome between the two arms [HR = 0.92 95% CI 0.69–1.22, *p* = 0.57] (Appendix A).

The summary HR for DFS showed that patients with *KRAS G12C* mutations had a higher risk of relapse compared to patients with other *KRAS* mutations [HR 1.49 95% CI 1.07–2.09, *p* < 0.0001], and significant heterogeneity was observed [I^2^ = 68%, *p* = 0.02] (Figure 2B). However, studies that compared *KRAS G12C* to *KRAS G12D* mutated tumors found a non-significant outcome [HR, 1.36, 95% CI, 0.59–3.15; *p* = 0.48] (Appendix A). Leave-one-out analysis revealed that none of the studies contributed to heterogeneity; however, the exclusion of any of the following studies led to a non-significant outcome: Finn et al., Jones et al., Nadal et al., and Villaruz et al. [40,46,47,49].

#### 3.3.2. KRAS G12C versus Wild-Type KRAS

Six studies with 4953 patients were included in the meta-analysis for OS [39,43,46,48,49,51]. Two studies with 2234 patients were included in the meta-analysis for DFS [46,49]. The summary HR for OS comparing *KRAS G12C* to *KRAS-WT* showed results favoring *KRAS-WT* [HR 1.42, 95% CI 1.10–1.84, *p* = 0.007], though significant heterogeneity was present [I^2^ = 68%, *p* = 0.008] (Figure 3A). Sensitivity analysis revealed consistent results. The meta-analysis for DFS comparing *KRAS G12C* with *KRAS-WT* showed similar survival outcomes between the two arms [HR 2.36, 95% CI 0.64–8.16, *p* = 0.19]; however, between-study heterogeneity was high [I^2^ = 93%, *p* = 0.0002] (Figure 3B).

### 3.4. Secondary Outcomes

We found that the proportion of patients with tumors expressing PD-L1 > 50% [OR 1.37 95% CI 1.11–1.70, *p* = 0.004] (Figure 4) were higher in the *mKRAS G12C* tumors compared to other *KRAS* mutations; however, tumors with neither negative (<1%) [OR 0.85 95% CI 0.70–1.04, *p* = 0.12] (Appendix A) nor with moderate PD-L1 expression (1–49%) [OR 0.94 95% CI 0.75–1.18, *p* = 0.59] (Appendix A) were associated with *mKRAS G12C*. Frequently encountered comutations did not show associations with either *mKRAS G12C* or other *KRAS* mutations [*TP53*: OR 0.96 95% CI 0.59–1.56, *p* = 0.88 (Appendix A); *STK11*: OR 1.01 95% CI 0.77–1.32, *p* = 0.93 (Appendix A)].

## 4. Discussion

Small molecule inhibitors of *KRAS G12C* mutant proteins finally broke the curse of the “undruggable” status of *KRAS* mutation based on the promising results of CodeBreak 100 and KRYSTAL-1 trials [25,52,53]. There is renewed interest in evaluating long-term oncological outcomes in patients with *mKRAS G12C* tumors [54]. Our systematic review and meta-analysis of sixteen retrospective studies comprising 10,153 patients showed that the *mKRAS-G12C* predicts poor survival in patients with NSCLC. The key findings of our study were as follows: First, compared to patients with other *KRAS* mutations, *mKRAS-G12C* predicted poor DFS. Second, *mKRAS-G12C* tumors were at a higher risk of all-cause mortality than *KRAS-WT* tumors despite similar DFS. Third, m*KRAS G12C* tumors were associated with high PD-L1 expression (>50%) compared with other *KRAS* mutations. Finally, *TP53* and *STK11* were not associated with either *mKRAS G12C* or other *KRAS* mutant tumors. Overall, *KRAS G12C* mutated tumors harbor poor prognosis; however, considerable between-study heterogeneity existed in most of these analyses.

In prior meta-analyses, the overall link between *mKRAS* and survival in NSCLC patients is relatively weak and primarily restricted to advanced disease [10,11]. Different mutation testing methodologies, variable patient selection criteria, and lack of stratification across stages further muddled the conclusions [55]. Furthermore, the downstream signaling of *KRAS* mutation subtypes uniquely alters tumor biology and thus may influence distinct clinical behavior that may not be apparent in examining *KRAS* mutations in toto [17,56]. Our analyses built on these lacunae by reporting the survival outcomes for the most common *KRAS* mutation subtype in NSCLC and showed consistent results across ethnicities and testing methodologies.

Surgically resected NSCLC are at a higher risk of disease relapse if associated with the *mKRAS G12C* than other *KRAS* mutations [47,49]. In contrast, tumors with *mKRAS G12C* and other *KRAS* mutations showed comparable OS rates in our study. In a study on early-stage NSCLC (*n* = 179), *mKRAS G12C* was associated with poor DFS (*p* = 0.006) compared with *KRAS-WT*; however, the statistical strength of this analysis dropped significantly for stage I patients, suggesting the influence of pathological stage on DFS [49]. Notwithstanding this, *KRAS G12C* inhibitors showed clinical efficacy in eradicating micrometastases and enhanced anti-tumor activity when combined with targeted agents [23]. Based on these results and our findings, adjuvant or neoadjuvant *KRAS G12C* inhibitor monotherapy or in combination with targeted agents in patients with early-stage *mKRAS G12C* tumors may open the door for exciting therapeutic paradigms.

*KRAS*-comutations and PD-L1 expression may also modify the behavior of *mKRAS* tumors [57,58]. Approximately 50% of *mKRAS* tumors exhibit co-mutations, of which *TP53* (35–50%), *STK11* (12–20%), and *KEAP1* (7–10%) constitute the majority [59,60]. *mKRAS G12C* tumors show similar concurrent genomic alterations, although they have also been associated with *ERBB2* amplification and *ERBB4* mutations [59]. *PTEN* and *PDGFRA* mutations are less commonly observed in *mKRAS G12C* tumors [59,61,62]. *KRAS*-comutations have been considered an adverse prognostic factor capable of predicting tumor progression and chemo-resistance [63,64,65,66]. Due to a lack of available data, we could not comment on the prognostic role of *KRAS*-comutations. However, data from clinical trials indicated that patients harboring *STK11* and *KRAS* co-mutations had a higher objective response rate (ORR) (63% vs. 33%), whereas *KEAP1* and *KRAS* co-mutation were associated with a lower ORR (20% vs. 44%) following *KRAS G12C* inhibitor therapy [52,53]. These genes need further investigation to evaluate their impact on prognosis and therapeutic response in patients with *mKRAS*. In this context, *KRAS G12C* inhibitors are being evaluated as a first-line therapeutic option in patients with NSCLC harboring *STK11* and *KRAS G12C* co-mutations (KRYSTAL-1 and CodeBreaK 201).

The degree of PD-L1 expression in tumor cells corresponds to the prognosis in patients with *mKRAS* tumors, and tumors with high PD-L1 expression were associated with a dismal prognosis [42,67,68]. Our study reported that high PD-L1 expression in tumor cells (>50%) was associated with *mKRAS G12C* tumors. In preclinical data, *KRAS G12C* direct inhibitors have been shown to upregulate a pro-inflammatory tumor microenvironment and increase anti-tumor T-cell activity [23]. Furthermore, *KRAS G12C* inhibitors have been shown to increase intratumoral chemokine concentrations, potentially increasing T-cell infiltration and enhancing immunosurveillance [23]. In CT-26 cell line models, *KRAS G12C* and anti-PD-1 combination therapy led to a synergistic response compared with either monotherapy [23]. In the CodeBreak 100 trial, 89.8% (53/59) of previously treated NSCLC patients with *mKRAS G12C* received anti-PD-1/PD-L1 therapy along with sotorasib and demonstrated an ORR of 32.2%, and a median PFS of 6.3 months [52]. Likewise, KRYSTAL-01 (NCT03785249), a phase1/2 multicohort study evaluating adagrasib (monotherapy or platinum-based chemotherapy/anti-PD-1/L1 combination), showed promising efficacy (42.9% ORR and median PFS of 6.5 months) in previously treated NSCLC harboring *mKRAS G12C* [53]. KRYSTAL-07 (NCT04613596) and CodeBreak 101 (NCT04185883) clinical trials are currently exploring anti-PD-1/*KRAS G12C* inhibitor combinations for advanced solid tumors.

We found that the frequency of *mKRAS-G12C* in the NSCLC and *mKRAS* cohorts reflected the real-world data that were previously reported, suggesting a representative population and thus more likely to reflect prognosis in clinical practice [69,70]. Acknowledging these findings cautiously is warranted. Most analyses involved high heterogeneity, which could not be attributed to an individual study and was presumably due to vast differences in patient characteristics. Publication bias was observed despite following expanded search criteria to include studies with either non-significant or negative results. The trim-and-fill method performs poorly in the presence of substantial between-study heterogeneity [71]. Approximately 49.7% of total patients had an advanced stage. Effect estimates from seven studies were unadjusted, which may influence the summary estimates, particularly by stage and systemic therapy. Lastly, this meta-analysis relied on the summary estimates from observational studies rather than individual patient data, which is considered a statistically superior method.

Trials like ADAURA and PACIFIC have provided sufficient evidence for the role of targeted agents in early-stage NSCLC. In addition, ANVIL (NCT02595944), PEARLS/Keynote-091 (NCT02504372), and IMpower010 (NCT02486718) trials are currently exploring the role of immune checkpoint inhibitors in the adjuvant setting [72,73]. However, the current literature lacks data to determine the prognostic and predictive role of *KRAS G12C* mutation in early-stage NSCLC; thus, further investigations are needed.

Our findings contribute to the evolving landscape of *KRAS* mutations in patients with NSCLC. Extrapolating these findings, the following recommendations may be made for clinical practice: 1) Inclusion of *KRAS* status, especially *KRAS G12C* mutation, as a routine test using a comprehensive molecular gene panel. Similarly, future studies should focus on 1) Evaluation of *KRAS G12C* inhibitors-anti-PD-1/PD-L1 inhibitor combination therapy in advanced NSCLC; 2) evaluation of adjuvant *KRAS G12C* inhibitor in surgically resected NSCLC with *mKRAS G12C*; and 3) assessment of the impact of *KRAS G12C*-comutations including *TP53*, *STK11*, *and KEAP1*, on prognosis.

## 5. Conclusions

Our meta-analysis on NSCLC found that tumors with *mKRAS G12C* were associated with worse DFS than tumors with other *KRAS* mutations and worse OS than tumors with *KRAS-WT*. The presence of significant heterogeneity and publication bias collectively undermines the validity of these findings. However, these outcomes reflect real-world prognoses and may be utilized in clinical practice to stratify high-risk patients and provide more effective therapeutic strategies for patients with NSCLC.

## Figures and Tables

**Figure 1 diagnostics-13-03043-f001:**
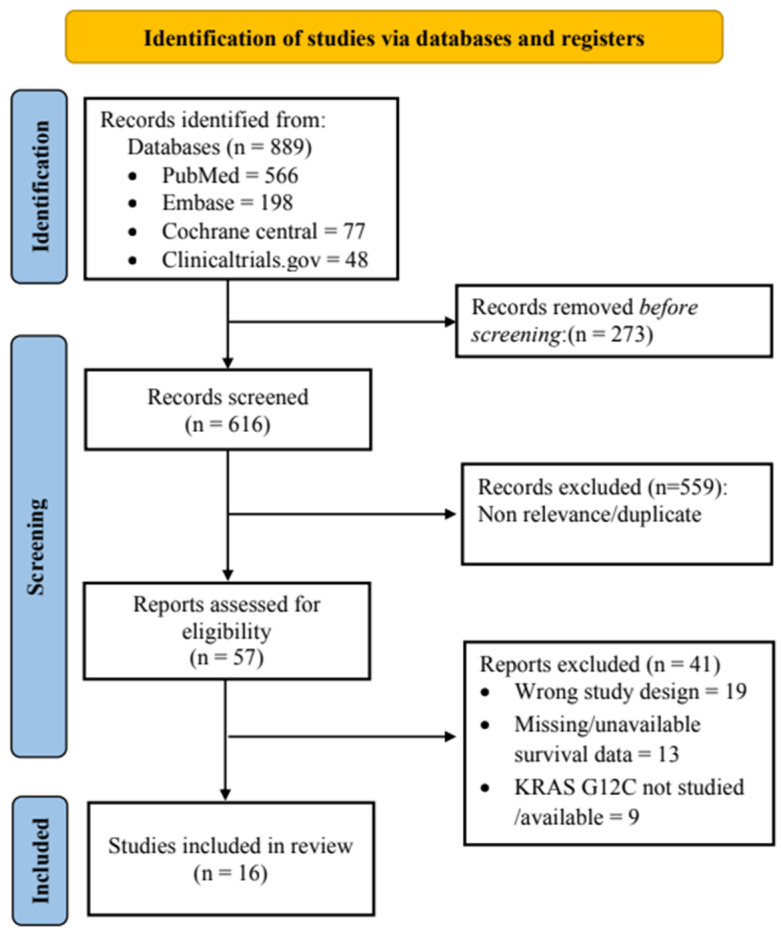
PRISMA flow diagram for the study selection process.

**Figure 2 diagnostics-13-03043-f002:**
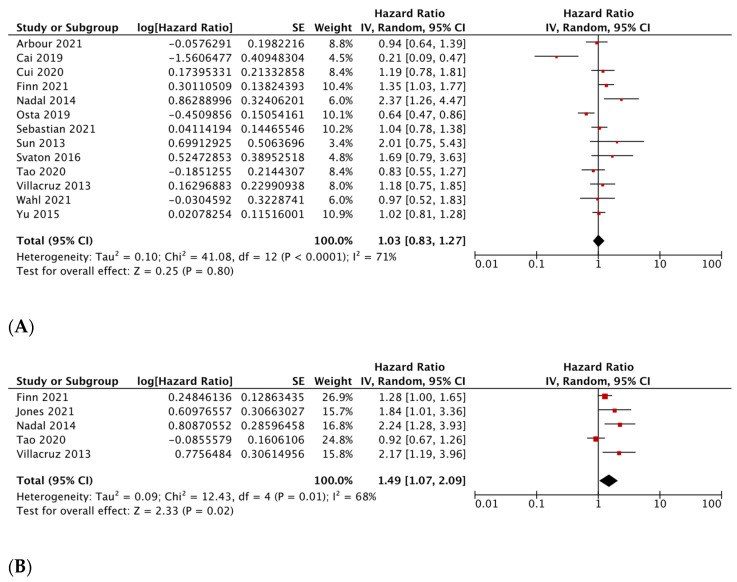
Association between *KRAS* mutation status and survival in non-small cell lung cancer. (**A**) Overall Survival for *KRAS G12C* mutant tumors versus other *KRAS* mutant tumors. Association between *KRAS* mutation status and survival in non-small cell lung cancer [36,37,38,39,40,41,42,43,44,45,46,47,48,49,50,51] (**B**). Disease-free survival for *KRAS G12C* mutant tumors versus other *KRAS* mutant tumors [40,42,46,47,49].

**Figure 3 diagnostics-13-03043-f003:**
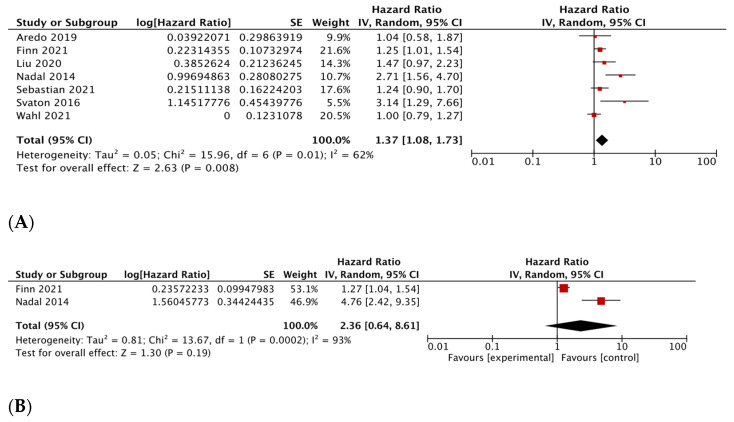
Association between KRAS mutation status and survival in non-small cell lung cancer, (**A**). Overall Survival for KRAS G12C mutant tumors versus wild-type KRAS tumors [37,39,43,46,48,49,51]; (**B**). Disease-free survival for KRAS G12C mutant tumors versus wild-type KRAS tumors [46,49].

**Figure 4 diagnostics-13-03043-f004:**
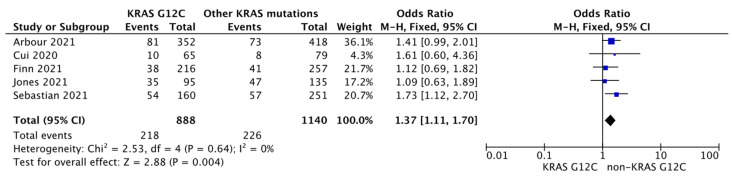
Association between KRAS status and PD-L1 in non-small cell lung cancer [36,45,46,47,51].

**Table 1 diagnostics-13-03043-t001:** Study characteristics.

Study	Source of Patients	No. of Patients (*n*)	Histology	Stage	Testing Method	KRAS G12C Mutation Frequency (%)	KRAS-Comutations (%)			Median Follow-Up Time (Months, Range)	Survival Parameters		NOS Score
							TP53	STK-11	KEAP-1		Outcome	Adjusted Variables	
Arbour, 2021 [36]	USA	772	NSCLC	IIIB-IV	NGS-MSK IMPACT assay	46	41.8	28.4	23.4	13.8	OS	NA	8
Aredo, 2019 [37]	USA	186	NSCLC	I-IV	NGS- STAMP assay	35	38.7	11.8	8.1	15	OS	Age, sex, smoking, stage, co-mutations, treatment modalities (localized and systemic)	9
Cai, 2019 [44]	China	84	NSCLC	IV	PCR-seq	28	29	NA	5	NA	OS	Age, sex, ECOG PS, smoking pack years, histology, KRAS comutations	6
Cui, 2020 [45]	Australia	346	NSCLC	IV	NGS-PMCC lung panel mutation analysis	19	NA	NA	NA	9	OS	NA	5
Finn, 2021 [46]	Multicentric (Europe)	2055	NSCLC	I-III	PCR-allele-specific multiplex test	10.5	NA	NA	NA	57.1 (44.3–72.3)	OS, DFS	Sex, ethnicity, smoking, age, Adj CT, Adj R.T., h/o cancer, ECOG PS, stage, primary tumor localization, tumor size, histology, surgery year, technique, and anatomy	9
Jones, 2021 [47]	USA	604	ADC	I-III	NGS- MSK-IMPACT	16	25	22	7	30 (IQR 21–40)	DFS	Tumor SUVmax, DLCO, LVI, VPI, STAS, stage	8
Liu, 2020 [48]	China	434	NSCLC	I-IV	NGS	9.6	NA	NA	NA	NA	OS	Age, sex, Smoking, histology, stage,	6
Nadal, 2014 [49]	USA	179	ADC	I-IV	PCR-seq	19.5	85	30.5	NA	95	OS, DFS	Age, sex, stage, smoking, adjuvant therapy, tumor differentiation,	9
Osta, 2019 [50]	USA	1655	ADC	IV	NGS	10.6	52	18	NA	25.8	OS	Age, sex, race, smoking, ECOG PS, prior surgical/R.T. treatment, systemic therapy, KRAS comutations	9
Sebastian, 2021 [51]	Germany	1039	NSCLC	IIIB-IV	NGS (75%)	15.4	NA	NA	NA	NA	OS	Age, sex, BMI, Charlton score, ECOG PS, PD-L1 expression	6
Sun, 2013 [38]	South Korea	304	NSCLC	IIIB-IV	PCR-seq	3	NA	NA	NA	30	OS	NA	6
Svaton, 2016 [39]	Czechia	129	NSCLC	IIIB-IV	PCR-seq	11.6	NA	NA	NA	NA	OS	NA	5
Tao, 2020 [42]	USA	254	ADC	I-IV	Pyrosequencing-based test (PyroMark Q24 System; Qiagen)	46.1	NA	NA	NA	17.2 (0.17–74.9)	OS, DFS	NA	6
Villacruz, 2013 [40]	USA	318	ADC	I-IV	PCR-seq	43.7	NA	NA	NA	24.3 (0–78)	OS, DFS	NA	6
Wahl, 2021 [43]	Norway	1117	ADC/ADSCC	I-IV	NGS	17	NA	NA	NA	52.7 (45.7–59.6)	OS	Age, sex, smoking, ECOG PS, Stage, Surgery, Curative/palliative CT/RT, TKI	9
Yu, 2015 [41]	USA	677	ADC	IV	Standard Sanger sequencing/	39	NA	NA	NA	17 (1–207)	OS	NA	6

HR: Hazard ratio, CI: Confidence Interval NGS: Next generation sequencing, NOS: Newcastle–Ottawa Scale, PCR-seq: Polymerase chain reaction-direct nucleotide sequencing, OS: Overall survival, DFS: Disease-free survival, P.S.: performance status, Adj: adjuvant, DLCO: diffusing capacity for carbon monoxide, LVI: lymphovascular invasion, VPI: visceral pleural invasion, STAS: spread through air spaces, BMI: Body mass index, ADC: adenocarcinoma, ADSCC: adenosquamous, CT: chemotherapy, R.T.: radiotherapy, TKI: tyrosine kinase inhibitor, NA: not available, STAMP: Solid Tumor Actionable Mutational Panel.

## Data Availability

The datasets generated during and/or analyzed during the current study are available in the Open Science Framework repository at 10.17605/OSF.IO/AUF2T without third-party permission.

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
