# Peer review of "Prognostic Role of KRAS G12C Mutation in Non-Small Cell Lung Cancer: A Systematic Review and Meta-Analysis"

_diagnostics, 2023, doi:10.3390/diagnostics13193043_

Round 1
Reviewer 1 Report
The authors demonstrated the relationship between KRAS G12C mutation (mKRAS G12C) and PD-L1 levels in NSCLC patients. The reviewer thinks this article is very interesting for Health Science and Oncology, however, it needs to clear in this article. To render the manuscript suitable for publication to Diagnostics, several corrections should be made.
Specific comments:
1. Figures 1-3, the quality was not good enough and should be improved.
2. All the Tables in this manuscript were very difficult to comprehend.
3. Please discuss more about the methodology for sample size estimation in the manuscript.
4. In the discussion section, please provide more information about age, gender, study area and stage of cancer in context with mKRAS G12C.
5. Please clarify the impact of PD-L1 expression levels on KRAS G12C and other KRAS mutations. Also, the relationship between PD-L1, TP53 and STK11 on KRAS G12C and other KRAS.
Minor editing of English language required.
Author Response
Dear Prof. Dr. Andreas Kjaer,
We submit a revision of our manuscript entitled " Prognostic Role of KRAS G12C Mutation in Non-Small Cell Lung Cancer- A Systematic Review and Meta-analysis" (Manuscript ID Diagnostics-2590728). The updated manuscript is attached, with revisions highlighted. We thank you and the reviewer for their insightful commentary and appreciate the opportunity to respond to the comments.
Thank you for your consideration.
Sincerely,
Dr. Durgesh Wankhede
General note: The changes in the manuscript have been highlighted, and Figure 1 and Table 1 have been modified as per the reviewers' comments; however, it has not been highlighted. The most up-to-date figure and table are re-attached.
Reviewer-1
- Figures 1-3, the quality was not good enough and should be improved.
Response: The authors thank the reviewer for highlighting the issues with the quality of the figures. We have updated the figures in the manuscript.
- All the Tables in this manuscript were very difficult to comprehend.
Response: The authors appreciate the reviewer’s comments and have updated the table. The table is also attached herewith:
|
Study |
Source of patients |
No. of patients (n)
|
Histology |
Stage |
Testing method |
KRAS G12C mutation frequency (%) |
KRAS-Comutations (%) |
Median follow-up time (months, range) |
Survival parameters |
NOS score |
|||
|
TP53 |
STK-11 |
KEAP-1 |
Outcome |
Adjusted variables |
|||||||||
|
Arbour, 2021 |
USA |
772 |
NSCLC |
IIIB-IV |
NGS-MSK IMPACT assay |
46 |
41.8 |
28.4 |
23.4 |
13.8 |
OS |
NA |
8 |
|
Aredo, 2019 |
USA |
186
|
NSCLC |
I-IV |
NGS-STAMP assay |
35 |
38.7 |
11.8 |
8.1 |
15 |
OS |
Age, sex, smoking, stage, co-mutations, treatment modalities (localized and systemic) |
9 |
|
Cai, 2019 |
China |
84 |
NSCLC |
IV |
PCR-seq |
28 |
29 |
NA |
5 |
NA |
OS |
Age, sex, ECOG PS, smoking pack years, histology, KRAS comutations |
6 |
|
Cui, 2020 |
Australia |
346 |
NSCLC |
IV |
NGS-PMCC lung panel mutation analysis |
19 |
NA |
NA |
NA |
9 |
OS |
NA |
5 |
|
Finn, 2021 |
Multicentric (Europe) |
2055
|
NSCLC |
I-III |
PCR-allele-specific multiplex test |
10.5 |
NA |
NA |
NA |
57.1 (44.3-72.3) |
OS, DFS |
Sex, ethnicity, smoking, age, Adj CT, Adj R.T., h/o cancer, ECOG PS, stage, primary tumor localization, tumor size, histology, surgery year, technique, and anatomy |
9 |
|
Jones, 2021 |
USA |
604
|
ADC |
I-III |
NGS- MSK-IMPACT |
16 |
25 |
22 |
7 |
30 (IQR 21-40) |
DFS |
Tumor SUVmax, DLCO, LVI, VPI, STAS, stage |
8 |
|
Liu, 2020 |
China |
434 |
NSCLC |
I-IV |
NGS |
9.6 |
NA |
NA |
NA |
NA |
OS |
Age, sex, Smoking, histology, stage, |
6 |
|
Nadal, 2014 |
USA |
179
|
ADC |
I-IV |
PCR-seq |
19.5 |
85 |
30.5 |
NA |
95 |
OS, DFS |
Age, sex, stage, smoking, adjuvant therapy, tumor differentiation, |
9 |
|
Osta, 2019 |
USA |
1655 |
ADC |
IV |
NGS |
10.6 |
52 |
18 |
NA |
25.8 |
OS |
Age, sex, race, smoking, ECOG PS, prior surgical/R.T. treatment, systemic therapy, KRAS comutations |
9 |
|
Sebastian, 2021 |
Germany |
1039 |
NSCLC |
IIIB-IV |
NGS (75%) |
15.4 |
NA |
NA |
NA |
NA |
OS |
Age, sex, BMI, Charlton score, ECOG PS, PD-L1 expression |
6 |
|
Sun, 2013 |
South Korea |
304 |
NSCLC |
IIIB-IV |
PCR-seq |
3 |
NA |
NA |
NA |
30 |
OS |
NA |
6 |
|
Svaton, 2016 |
Czechia |
129 |
NSCLC |
IIIB-IV |
PCR-seq |
11.6 |
NA |
NA |
NA |
NA |
OS |
NA |
5 |
|
Tao, 2020 |
USA |
254 |
ADC |
I-IV |
Pyrosequencing-based test (PyroMark Q24 System; Qiagen) |
46.1 |
NA |
NA |
NA |
17.2 (0.17-74.9) |
OS, DFS |
NA |
6 |
|
Villacruz, 2013 |
USA |
318 |
ADC |
I-IV |
PCR-seq |
43.7 |
NA |
NA |
NA |
24.3 (0-78) |
OS, DFS |
NA |
6 |
|
Wahl, 2021 |
Norway |
1117 |
ADC/ADSCC |
I-IV |
NGS |
17 |
NA |
NA |
NA |
52.7 (45.7-59.6) |
OS |
Age, sex, smoking, ECOG PS, Stage, Surgery, Curative/palliative CT/RT, TKI |
9 |
|
Yu, 2015 |
USA |
677 |
ADC |
IV |
Standard Sanger sequencing/ |
39 |
NA |
NA |
NA |
17 (1-207) |
OS |
NA |
6 |
- Please discuss more about the methodology for sample size estimation in the manuscript.
Response: The authors would like to thank the reviewer for highlighting this essential aspect of this study. Each study provided sample sizes for individual comparative groups and have been extracted from these individual studies. The information is provided on pages 4-5, lines 171-173.
- In the discussion section, please provide more information about age, gender, study area, and stage of cancer in context with mKRAS G12C.
Response: The author thank the reviewer for their comment. However, since individual data for each of these variables (age and gender) were not available in all studies, formal analysis was not carried out. In addition, there were two control groups (other KRAS mutations and KRAS-WT) which precluded a definitive estimation of the demographic characteristics of mKRAS G12C patients.
All studies were retrospective (Page 9, lines 280-284) and 49.7% of total participants had advanced-stage disease (IIIb-IV) (Page 10, lines 352-354). This information has been updated in the revised manuscript.
- Please clarify the impact of PD-L1 expression levels on KRAS G12C and other KRAS mutations. Also, the relationship between PD-L1, TP53, and STK11 on KRAS G12C and other KRAS.
Response: The authors thank the reviewer for their comment and have provided this information on these salient variables in the revised manuscript (Page 10, lines 312-345) as follows:
“KRAS-comutations and PD-L1 expression may also modify the behavior of mKRAS tumors [57,58]. Approximately 50% of mKRAS tumors exhibit co-mutations, of which TP53 (35-50%), STK11 (12-20%), and KEAP1 (7-10%) constitute the majority[59,60]. mKRAS G12C tumors show similar concurrent genomic alterations, although they have also been associated with ERBB2 amplification and ERBB4 mutations[59]. PTEN and PDGFRA mutations are less commonly observed in mKRAS G12C tumors[59,61,62]. KRAS-comutations have been considered an adverse prognostic factor capable of predicting tumor progression and chemo-resistance[63–66]. Due to a lack of available data, we could not comment on the prognostic role of KRAS-comutations. However, data from clinical trials indicated that patients harboring STK11 and KRAS co-mutations had a higher objective response rate (ORR) (63% vs. 33%), whereas KEAP1 and KRAS co-mutation were associated with a lower ORR (20% vs. 44%) following KRAS G12C inhibitor therapy[52,53]. These genes need further investigation to evaluate their impact on prognosis and therapeutic response in patients with mKRAS. In this context, KRAS G12C inhibitors are being evaluated as a first-line therapeutic option in patients with NSCLC harboring STK11 and KRAS G12C co-mutations (KRYSTAL-1 and CodeBreaK 201).”
“The degree of PD-L1 expression in tumor cells corresponds to the prognosis in patients with mKRAS tumors, and tumors with high PD-L1 expression was associated with dismal prognosis[42,67,68]. Our study reported that high PD-L1 expression in tumor cells (>50%) was associated with mKRAS G12C tumors. In preclinical data, KRAS G12C direct inhibitors have been shown to upregulate a pro-inflammatory tumor microenvironment and increase anti-tumor T-cell activity[23]. Furthermore, KRAS G12C inhibitors have been shown to increase intratumoral chemokine concentrations, potentially increasing T-cell infiltration and enhancing immunosurveillance[23]. In CT-26 cell line models, KRAS G12C and anti-PD-1 combination therapy led to a synergistic response compared with either monotherapy [23]. In the CodeBreak 100 trial, 89.8% (53/59) of previously treated NSCLC patients with mKRAS G12C received anti-PD-1/PD-L1 therapy along with sotorasib and demonstrated an ORR of 32.2%, and a median PFS of 6.3 months[52]. Likewise, KRYSTAL-01 (NCT03785249), a phase1/2 multicohort study evaluating adagrasib (monotherapy or platinum-based chemotherapy/anti-PD-1/L1 combination), showed promising efficacy (42.9% ORR and median PFS of 6.5 months) in previously treated NSCLC harboring mKRAS G12C[53]. KRYSTAL-07 (NCT04613596) and CodeBreak 101 (NCT04185883) clinical trials are currently exploring anti-PD-1/KRAS G12C inhibitor combinations for advanced solid tumors. “

Reviewer 2 Report
The article entitled "Prognostic Role of KRAS G12C Mutation in Non-Small Cell 2 Lung Cancer- A Systematic Review and Meta-analysis" provides a systematic review of the prognostic value of mutations in the KRAS gene in NSCLC. This is undoubtedly a very interesting review that addresses a highly topical issue of great therapeutic importance, both because of the current drugs that act on KRAS, such as sotorasib or adagrasib, and because of their impact on immunotherapy. The authors present a perfectly written article that meets the objectives presented and the title that one can read. The main flaw is the existence of other articles with similar characteristics such as Judd J et al (Cancer Mol Cancer Ther. 2021;20(12):2577-2584), however, it provides some originality as well as an update of already published knowledge.
The article is perfectly structured with language that needs no editing. The methodology is impeccable, meeting all the PRISMA criteria required for a systematic review. The figures and tables shown are those required for the article and do not need any changes. The references are mostly up to date and allow a correct bibliographic search if necessary or if you want to know more about the subject. In my view, there are no doubts about the results, discussion or conclusions. They are correct and do not need major changes except for some minor points. Therefore, the systematic review carried out by the authors is impeccable and I only add a number of minor changes that I believe would improve the article:
Abstract: add the PROSPERO registry.
Keywords: replace "lung cancer" with "non-small cell lung cancer". Add "systematic review".
Introduction: comment on which patients the KRAS mutations appear in. Some of them are more associated with smokers than others. It is important to take this into account.
Introduction: indicate that ESMO currently recommends it for second-line treatment and depending on which clinical trials this approval is given.
Materials and methods: put the inclusion and exclusion criteria at different points.
Results: the arrows in the flowchart are not correctly displayed.
Results: I would put supplementary figure 10 in the main text.
Author Response
Dear Prof. Dr. Andreas Kjaer,
We submit a revision of our manuscript entitled " Prognostic Role of KRAS G12C Mutation in Non-Small Cell Lung Cancer- A Systematic Review and Meta-analysis" (Manuscript ID Diagnostics-2590728). The updated manuscript is attached, with revisions highlighted. We thank you and the reviewer for their insightful commentary and appreciate the opportunity to respond to the comments.
Thank you for your consideration.
Sincerely,
Dr. Durgesh Wankhede
General note: The changes in the manuscript have been highlighted, and Figure 1 and Table 1 have been modified as per the reviewers' comments; however, it has not been highlighted. The most up-to-date figure and table are re-attached.
Reviewer-2
- Abstract: add the PROSPERO registry.
Response: The authors appreciate the reviewer’s comments and have added the information in the revised manuscript. (Page 1, line 22)
- Keywords: replace "lung cancer" with "non-small cell lung cancer". Add "systematic review".
Response: The authors appreciate the reviewer’s comments and have added the information in the revised manuscript. (Page 1, lines 36)
- Introduction: comment on which patients the KRAS mutations appear in. Some of them are more associated with smokers than others. It is important to take this into account.
Response: The authors thank the reviewer’s comments and have added the information in the revised manuscript. (page 1, lines 45-46, page 2, 57-59)
- Introduction: indicate that ESMO currently recommends it for second-line treatment and depending on which clinical trials this approval is given.
Response: The authors thank the reviewer’s comments and have added the information in the revised manuscript. (Page 2, lines 69-71)
“Similarly, based on the results of the CodeBreak200 trial (n = 345), ESMO guidelines recommend sotorasib as second-line therapy for advanced NSCLC patients with mKRAS G12C[26,27].”
- Materials and methods: put the inclusion and exclusion criteria at different points.
Response: The authors thank the reviewer’s comments and have formatted the information in the revised manuscript. (Pages 2-3, lines 93-108)
- Results: the arrows in the flowchart are not correctly displayed.
Response: The authors appreciate the reviewer’s comments and have revised the figure.
7. Results: I would put supplementary figure 10 in the main text.
Response: The authors appreciate the reviewer’s comments and have added the figure in the revised manuscript.